# Integration of Genomic Profiling and Organoid Development in Precision Oncology

**DOI:** 10.3390/ijms23010216

**Published:** 2021-12-25

**Authors:** Hyunho Yoon, Sanghoon Lee

**Affiliations:** 1Department of Medical and Biological Sciences, The Catholic University of Korea, Bucheon 14662, Korea; hyoon@catholic.ac.kr; 2Department of Obstetrics and Gynecology, Korea University College of Medicine, 73, Inchon-ro, Seongbuk-gu, Seoul 02841, Korea

**Keywords:** precision oncology, next-generation sequencing, organoids, genome profiling, breast cancer, ovarian cancer

## Abstract

Precision oncology involves an innovative personalized treatment strategy for each cancer patient that provides strategies and options for cancer treatment. Currently, personalized cancer medicine is primarily based on molecular matching. Next-generation sequencing and related technologies, such as single-cell whole-transcriptome sequencing, enable the accurate elucidation of the genetic landscape in individual cancer patients and consequently provide clinical benefits. Furthermore, advances in cancer organoid models that represent genetic variations and mutations in individual cancer patients have direct and important clinical implications in precision oncology. This review aimed to discuss recent advances, clinical potential, and limitations of genomic profiling and the use of organoids in breast and ovarian cancer. We also discuss the integration of genomic profiling and organoid models for applications in cancer precision medicine.

## 1. Introduction

Genetic variations and mutations generally increase the risk of cancer. *BRCA1* and *BRCA2* are known genetic risk factors for breast and ovarian cancers. These gene mutations are reported in 3% of breast cancer and 10–15% of ovarian cancers [1,2], and are strongly associated with cancer development. A family history of multiple breast or ovarian cancer accounts for about 15% of all breast cancer cases [3], suggesting that an integrated investigation of breast and ovarian cancers should lead to a better understanding of these cancers.

In contrast to the one-drug-fits-all model, the precision medicine approach aims to provide clinical benefits to patients based on their individual molecular matching [4,5]. Cancer treatment is a major topic in precision medicine because cancer has many difficulties that cannot be treated with a single drug. In the past two decades, genotyping and genomics have become an integral part of the standard treatment for various cancers, including breast and ovarian cancer [6]. Several studies have shown that next-generation sequencing (NGS)-based approaches provide genetic information obtained from both tumor and stromal cells of cancer patient tissue, including mutational status and gene expression patterns of cancer cells, leading to sequence-matching therapy and improving overall drug response and survival rates of cancer patients [7]. 

Thus, the elucidation of the precise and detailed genetic landscape of cancer patients may provide a clinical advantage in precision oncology [8,9,10]. The complete genome or target regions of DNA or RNA can be sequenced using NGS, an accurate and rapid method to obtain detailed genetic information of cancer patients. Advances in the bioinformatic analysis of NGS data have enabled the precise identification of genetic alterations, including single nucleotide variants, gene fusions, and somatic mutations [11]. This technology performs a variety of applications, including whole-genome sequencing, whole-exome sequencing, and whole-transcriptome sequencing (RNA-seq), to address the molecular landscape of the cancer genome. However, tumors are composed of many distinct cell types, and they are characterized by distinct genetic changes by communicating with various cells, such as immune cells and cancer-associated fibroblasts (CAFs) [12,13]. Thus, NGS-based single-cell technologies, such as single-cell RNA sequencing (scRNA-seq), single-cell assay for transposase-accessible chromatin-sequencing (scATAC-seq), and single-cell chromatin immunoprecipitation sequencing (scChIP-seq) have been developed and provide a broad high-resolution view of individual cancer patients at the single-cell level, opening up innovative therapeutic options for precision oncology [14,15,16,17].

Two-dimensional (2D) cancer cell lines are useful for studying cancer biology in vitro. However, the currently available cancer cell lines are associated with limitations, including the lack of tumor heterogeneity and tumor stroma. Organoid models are an innovative engineering approach that reflects aberrant genomic variations in cancer patients, representing the genetic and functional properties of cancer cells [18,19,20]. Thus, organoids have been employed to establish practical models of tumor development, initiation, and metastasis, as well as to examine the efficacy of various therapeutic agents for several cancers, including breast and ovarian cancer. In addition, advanced organoid models (i.e., organoids-on-chip) may have potential clinical applications in discovering novel drugs, determining therapeutic strategies, and developing personalized cancer treatment [21,22].

Both cancer organoid models and genome profiling using NGS techniques are powerful tools to find better strategies for cancer treatment. In this review, we discuss the integration of genomic profiling and organoid models in breast and ovarian cancer to provide better ideas for developing personalized cancer therapies.

## 2. Single-Cell Sequencing-Matched Cancer Treatment

Cancer is a disease characterized by extreme genetic instability and variation. Various approaches for dealing with genomic information are being actively developed. As a result, NGS-based RNA-seq provides many cancer treatment targets and strategies by more easily and quickly acquiring the genetic landscape of cancer cells. For example, whole-transcriptome sequencing has uncovered genomic abnormalities and complexity in various tumors, providing therapeutic targets and better strategies for cancer treatment [23]. However, since bulk RNA-seq-related studies analyze the average value of all constituent cells of cancer tissues, there is a limit to the accurate single-cell unit analysis. Cancer consists of a diverse population of cells with distinct genotypes and phenotypes. Therefore, to accurately understand this heterogeneous disease, it is necessary to analyze each distinct cell population. Recent advanced single-cell sequencing technologies have allowed analysis of the phenotype and genotype of different single cells, providing the degree and prevalence of intratumoral and intertumoral heterogeneity (Figure 1). For example, the scRNA-seq technology has provided a high-resolution view of intercellular differences, including genetic mutations, gene expression patterns, developmental hierarchy, and epigenetic modifications, which facilitates an enhanced understanding of the function of an individual tumor cell using its transcriptional output [24,25,26]. Therefore, precision oncology based on single-cell sequencing-matched results offers significant clinical advantages for cancer treatment [27]. Here, we will discuss single-cell sequencing-matched therapies in ovarian and breast cancer (Table 1).

### 2.1. Ovarian Cancer

Ovarian cancer is one of the most malignant and fatal gynecological tumors. High-grade serous ovarian cancer (HGSOC) is a critical histological type of ovarian cancer [2,28,29]. Germline mutations in *BRCA1* and *BRCA2*, which are reported in 10–15% of ovarian cancers, play an essential role as genetic risk factors for cancer development [30]. BRCA mutations are reported to increase the risk of developing HGSOC. In addition, *BRCA1* mutations are strongly associated with poor clinical outcomes in HGSOC patients [31,32]. In 2011, The Cancer Genome Atlas (TCGA), using NGS and microarray data analyzed from HGSOC patients, revealed four subtypes of HGSOC based on gene expression: mesenchymal, immune response, proliferation, and differentiation [2,33]. Using NGS data, Murakami et al. established a genetically consistent histopathological classification system for HGSOC. The authors reported that the mesenchymal subtype was the most aggressive and sensitive to taxanes, such as paclitaxel [33].

Ovarian cancer, like other tumors, is characterized by genetic heterogeneity. HGSOC is known to be associated with high rates of genomic instability and TP53 mutations [34]. The intratumoral genetic heterogeneity of HGSOC was revealed by loss of heterozygosity and comparative genomic hybridization, leading to implications for the molecular diagnosis of ovarian cancer [35,36,37,38,39]. In addition, recent studies using scRNA-seq analysis have further revealed high-resolution views of the molecular environment of individual cells and distinct cell subtypes that characterize the intratumoral heterogeneity of ovarian cancer tissues [40,41,42,43,44]. scRNA-seq data from multiple ovarian cancer ascites provided gene expression profiles of individual cells, identified the immune and stromal cells, and characterized their interactions and contributions to cancer pathogenesis and resistance to chemotherapy [45,46]. Izar et al. performed scRNA-seq using a set of 22 ascite samples from 11 patients, two primary HGSOC tumors, and three patient ascite-derived xenograft models. The authors analyzed the expression levels of immune-related genes and reported that JAK/STAT signaling contributes to the inflammatory programming of HGSOC cells [47]. Subsequently, 15 compounds targeting the JAK/STAT pathway or its effectors were screened. JSI-124 (cucurbitacin I), which targets the STAT3 signaling pathway, increases the cancer cell death rate and extent of disease response in vivo [47,48], suggesting that the JAK/STAT pathway could be a promising target for HGSOC treatment. In another recent study, scRNA-seq was performed using ovarian cancer, normal ovarian, and embryonic tissues to investigate heterogeneity [49]. This study showed a comparison of gene expression profiles between ovarian cancer and embryonic tissues. Interestingly, PEG10+ clusters have been identified in both ovarian cancer and embryonic tissues, which modulate cancer stem cell activity and drug resistance in ovarian cancer cells [49]. This study suggested that the PEG10-mediated cancer embryo population could be a therapeutic target for ovarian cancer.

Tumor-infiltrating lymphocytes (TILs), such as B cells, T cells, macrophages, and dendritic cells, are often recruited in many solid tumor tissues, which is required for response to cancer immunotherapy. In general, observing large numbers of TILs in cancer patients significantly improved patient outcomes in many types of solid tumors, including breast and ovarian cancers [50,51]. Therefore, it is essential that scRNA-seq identifies tumor composition, including immune cells, to provide cellular heterogeneity at single-cell resolution. An ovarian cancer research group analyzed scRNA-seq data of 9885 cells isolated from the omentum in six ovarian cancer patients [52]. As a result, nine major cell types were identified, including cancer, stromal, and immune cells, and high T cell infiltration significantly enhanced the antitumor response. These findings suggest that scRNA-seq studies provide essential insights to identify specific immune cell clustering to improve immune responses within ovarian tumors. In another study that performed scRNA-seq analysis in HGSOC, chemo-resistant cell population was identified by functional classification of single cells [40]. Even though known chemo-resistant signature genes in ovarian cancer were not expressed much in the chemo-resistant cell population, the population contributed to chemo-resistance by producing cancer stem genes, such as CD44, MYD88, and ALDH1 [53]. This study suggested therapeutic options targeting the chemo-resistant cell population to increase therapeutic effects in ovarian cancer. Taken together, advanced single-cell genome profiling to identify intratumoral heterogeneity may provide new perspectives for understanding cancer progression, and improve the efficacy of ovarian cancer treatment.

### 2.2. Breast Cancer

Breast cancer is the most common cancer in women. Strategies targeting human EGFR2 (HER2), a tyrosine receptor kinase, have been widely and successfully used to treat HER2-positive breast cancer [54,55,56]. HER2 is reported to be upregulated in 20–25% of breast cancers, which are aggressive tumors [57,58,59]. Herceptin, a monoclonal anti-HER2 antibody, is a Food and Drug Administration (FDA)-approved treatment for HER2-positive breast cancer. Despite the remarkable efficacy of Herceptin against breast cancer, the disease still causes death in many women due to relapse and the development of drug resistance [60]. Some studies have elucidated the critical mechanisms involved in the resistance to therapy, such as the activation of alternative signaling pathways. For example, the overexpression of receptors, such as hepatocyte growth factor receptor (HGFR), CXCR4, and integrins, may contribute to the resistance to Herceptin by activating cytoplasmic signaling pathways [61,62,63]. 

Resistance-associated receptors are mainly activated by the interaction between tumor cells and other types of cells. Hence, we hypothesize that this interaction may be strongly associated with tumor heterogeneity. Single-cell transcriptome profiling elucidated the components, including clonal evolution and stromal and immune cells, of the heterogeneous breast cancer cells [64,65,66,67,68]. Gao et al. developed a novel single-cell approach called nanogrid single-nuclear RNA sequencing, which combines nanogrid technology with scRNA-seq. The authors used state-of-the-art tools to compare breast tumor tissue (416 nuclei) with normal breast tissue (380 nuclei). The analysis revealed the heterogenous phenotypic profiles of breast cancer related to angiogenesis, cell proliferation, and cancer stemness [69]. Triple-negative breast cancer (TNBC), which is the most aggressive breast cancer, does not express estrogen receptor, progesterone receptor, and HER2. Thus, TNBCs are associated with fewer therapeutic targets. As EGFR is often upregulated in TNBC, it is considered an important therapeutic target for treating TNBC. However, EGFR-targeted therapies (such as gefitinib and cetuximab) are associated with modest but variable clinical benefits (1.7–38.7%) [70,71,72]. Gefitinib is approved for first-line treatment of advanced non-small-cell lung cancer (NSCLC) with EGFR mutation. Cetuximab is a chimeric monoclonal antibody that binds to and inhibits EGFR [73]. While cetuximab treatment has failed in colorectal cancer with EGFR mutation, gefitinib showed median overall survival to be 37 months in advanced EGFR-mutant NSCLC [74,75]. Although EGFR mutation plays an important role that can be directly linked to the therapeutic effect of the drugs, accurate information of EGFR mutation was not available from breast cancer research results. To address the varying rates of efficacy of anti-EGFR therapeutics, Savage et al. performed scRNA-seq analysis using single cells from clinical TNBC-derived xenografts. Detailed genomic profiling demonstrated that TNBC exhibits heterologous EGFR expression, and high expression levels of EGFR were strongly associated with cancer stemness and sensitivity to gefitinib [76]. This study suggested that a specific population associated with cancer stemness and drug resistance should be targeted for the treatment of breast cancer.

In TNBC, since a large number of TILs were recruited into cancer tissues [77,78], anti-cancer immunotherapy was predicted to be used in a promising manner. Indeed, the neutralizing PD-L1 antibody, atezolizumab, improved patient survival when combined with nab-paclitaxel as a first-line treatment in PD-L1+ metastatic TNBC [79]. However, the detailed mode of action remains unclear. Thus, Bassez et al. performed single-cell transcriptome analysis using 40 TNBC patients with neoadjuvant anti-PD1 to identify a subset of signature genes that respond to the neutralizing PD-L1 antibody. As a result, CCR2+ or MMP9+ macrophage and dendritic cells positively increased with T cell expansion, leading to changes of immune-related microenvironment under anti-cancer immunotherapy [80]. These results demonstrated that the signature genes analyzed in the single-cell transcriptome tool can provide insight into the therapeutic targeting of candidate proteins for synergistic effect with anti-PD1 [80], suggesting that scRNA-seq can be used to study the synergistic effect of anti-tumor immune responses in breast cancer.

In cancer stem cell (CSC) biology, single-cell transcriptome approaches have been widely used to identify putative CSC biomarkers involved in the Notch, Hedgehog, and Wnt signaling pathways, leading to induction of self-renewal and drug resistance in cancer cells [81,82,83]. Consequently, a Notch signaling inhibitor (i.e., γ-secretase inhibitor) was evaluated in clinical study and these studies were extended into phase I/II study of MK-0752 (γ-secretase inhibitor), followed by docetaxel in advanced or metastatic breast cancer [84]. These findings suggest that single-cell transcriptome approaches can serve as therapeutic targets to treat breast cancer. Taken together, sequencing-matched therapies should be applied to breast cancer to improve treatment outcomes, and deep genomic profiling at the single-cell level may broaden treatment options for individual breast cancer patients.

## 3. Advanced Genome Profiling Combined with the Use of Cancer Organoids

Traditional cancer cell lines, which are established from primary and metastatic tumor lesions, are immortalized and cultured as 2D monolayers. More recently, several cell lines of patient-derived xenograft (PDX) and chemotherapy-resistant cancers have been developed for cancer research. These cancer cell lines are valuable for investigating the mechanisms underlying cancer-specific cytoplasmic signaling, testing drug efficacy in vitro and in vivo, and identifying potential cancer therapeutics. However, cancer cell lines do not mimic pathological conditions, such as the heterogeneous environment and genetic features of the tumors, which leads to discrepancies between experimental and clinical results [85,86]. Cancer organoids are in vitro 3D models that represent the genetic and molecular diversities of cancer cells and the complexity of the tumor microenvironment (TME) [19]. Thus, cancer organoids can enhance our understanding of cancer biology and are viable alternatives to the traditional cancer cell lines.

Using valuable cancer patient samples, advances in single-cell transcriptomics have recapitulated the molecular diversity, cellular heterogeneity, and genetic landscape of patient tumors. However, due to the difficulty of accessing patient tissues, organoid models are becoming a necessary alternative material for single-cell sequencing approaches. Indeed, integrated analysis of cancer patient organoid models and scRNA-seq further facilitated tumor heterogeneity and detailed genetic information, leading to therapeutic options for personalized cancer treatment. As an example, in glioblastomas, the resection of primary samples yielded limited information on the interactions between invasive tumor cells and non-tumorous brain cells owing to the limitations associated with isolating neoplastic cells from the tumor periphery [87]. Krieger et al. performed single-cell transcriptome analysis with glioblastoma organoids and demonstrated cellular interactions of invasive glioblastoma cells that are regulated by transcriptional changes [88]. Recently, Norkin et al. developed a targeted organoid sequencing method (TORNADO-seq that provides an efficient and high-throughput drug screening platform in colorectal cancer organoids [89]. TORNADO-seq uses intestinal and cancer organoids to identify various targetable signatures of cellular status and signaling pathways in individual colorectal cancer cells associated with cellular differentiation status [89]. Therefore, advanced genomic profiling combined with cancer organoid models can provide useful insights for understanding cancer biology. Here, we will discuss more on the clinical benefits of advanced methods in ovarian and breast cancer (Table 2). 

### 3.1. Ovarian Cancer

Cancer organoids can reveal the pathological state of the original tumor. Hence, cancer organoids are widely used as preclinical models for translational cancer studies. For ovarian cancer, the limited in vitro 3D experimental models make it difficult to find better ovarian cancer treatment strategies. To overcome the shortcomings, ovarian cancer organoid models that maintain the heterogeneity of tumors and the genomic landscape of their parental tumors are being developed [90]. Kopper et al. have established 56 ovarian cancer organoid lines from 32 patients that maintain the genomic features of corresponding tumors [91]. scRNA-seq data performed with these organoids and patient samples showed that organoids maintained tumor heterogeneity during long-term expansion. In addition, in vivo drug sensitivity analysis using ovarian cancer organoids suggested potential applications for the gold standard platinum-based chemotherapy [91]. However, long-term maintenance of cancer organoids is difficult due to low survival rates and morphological changes, which lead to tumor heterogeneity and loss of genomic features of the parental tumor. Thus, short-term patient-derived organoids (PDO) have been tested for the identification of druggable targets for ovarian cancer. Hill et al. have established 33 HGSOC organoids maintaining DNA repair defects derived from 22 patients [92]. Based upon genomic analyses, up to 50% of HGSOC altered with DNA damage response genes, which can be a prediction of HGSOC [41]. Short-term patient-derived HGSOC organoids that maintained the features of DNA repair genes were tested for the identification of targetable DNA damage repair defects, which accurately predicted ATR (Ataxia telangiectasia and Rad3)-related inhibitor and PARP (Poly [ADP-ribose] polymerase 1) inhibitor resistance and prexasertib, carboplatin, and gemcitabine sensitivity [92]. Similarly, short-term PDOs were established from each histologic subtype of ovarian tumor (three HGSOC, one clear cell, three endometrioid), with a success rate of 80% [93]. Targeted capture sequencing of 1053 cancer-related genes revealed that the ovarian cancer organoids harbored the characteristics of histological subtypes, pivotal DNA variants, similar copy number variation (CNV) profiles, and mutation status of parental tumor, which were evaluated for drug screening with 23 FDA-approved chemotherapy drugs [93]. This research showed that advanced approaches of ovarian cancer organoids with sequencing-matched genome profiling efficiently serve as a platform for drug screening in translational research and precision medicine. 

### 3.2. Breast Cancer

Organoids have been established for primary and metastatic breast cancer tissues. Breast cancer organoids are reported to maintain distinct cell populations and exhibit cellular heterogeneity and are associated molecular subtypes of their parent cancer cells [94,95,96,97,98]. PLK4, which plays an essential role in EMT, cell proliferation, and centriole replication during cell cycle progression, is upregulated in breast cancer [99,100,101]. Previous studies have reported that PLK4 is a potential therapeutic target for epithelial cancer. The rates of anti-cancer efficacy of various small-molecule inhibitors (such as YLT11, YLZ-F5, and CFI-400945) of PLK4 have been recently investigated. In particular, preclinical studies using organoids derived from patients with breast cancer revealed that CFI-400945 alone or in combination with radiation treatment exerted synergistic growth-inhibitory effects on TNBC [100]. Interestingly, organoids can also contain immune cells, such as B cells, T cells, natural killer cells, and macrophages [102]. Mammary ductal epithelial organoids contain a small percentage of Vδ1/2 T cells (0.42–1.27%), the prevalence of which increases upon treatment with amino bisphosphonate drugs [103]. Bisphosphonate drugs are critical for maintaining and developing ductal epithelial organoids. Organoid-derived Vδ2(+) T cells exerted growth-inhibitory effects against breast cancer cells through the production of interferon gamma [103]. This indicated that organoid-mediated immune cells can have potential applications in immunotherapeutic approaches to treat cancer. In order to use cancer organoids effectively, Sachs et al. have established a living biobank of breast cancer organoids from ~150 breast cancer patients [94]. The breast cancer organoids have been identified as retaining the genomic landscape (e.g., mutational signatures) as well as characteristic histological features of tumor cells. The HER signaling pathway was selected by organoid RNA-seq analysis that identified putative molecular targets and pathways for breast cancer treatment. The organoids were sensitive to drugs (e.g., afatinib and pictilisib) blocking the HER signaling pathway in vitro and in vivo, proposing breast cancer organoids as a promising pre-clinical model. These findings also indicate that genomic profiling and the use of cancer organoids are promising tools for personalized cancer treatment.

## 4. Organoids on a Chip

Cancer organoids are promising models for improving our understanding of cancer biology and performing high-throughput personalized drug screening. However, limitations of cancer organoids include the absence of cancerous stroma such as fibroblasts, blood vessels, and immune cells. To address this limitation, various systemic organoids that mimic the parental stroma have been generated. Recently, the air–liquid interface patient-derived PDO that retains tumor immune cells, including TILs and fibroblasts, was reported [102]. RNA-seq with individual organoid single cells revealed the immune diversity and T cell receptor repertoire of the original tumor biopsies. PD-1/PD-L1 blockade expands TILs within mouse tumor organoids, and human organoid TILs functionally recapitulate PD-1-dependent immune checkpoints [102]. Similarly, co-culturing with endothelial cells or progenitor cells has been successfully used as a model of vascularized organoids mimicking brain, liver, and kidney microenvironments [104,105,106]. As an example, a tumor-vasculature-on-a-chip model allowing co-culture of human umbilical vein endothelial cells (HUVEC) with SKOV3 spheroids recapitulated endothelial barrier function, indicating that it facilitates better prediction of the transport efficacy for cancer organoid models [107]. A microfluidic model has been developed to study the immune response and contains breast cancer spheroids in a 3D extracellular matrix and two lateral lumens lined with endothelial cells. This system allowed the antibody to perfuse through the lateral lumen leaking from the vessels and rapidly diffuse through the stroma [108]. Furthermore, microfluidic chips with nanoparticle have been established as systemic vascularized organoid models with self-derived vascular networks [109]. The microfluidic chip was developed by nanoparticle functionalized bioinks, which control distinct growth factor release profiles. The system enhanced vessel infiltration and angiogenesis by distinct VEGF gradient [109]. Taken together, organoids-on-a-chip platforms can mimic the structural and functional characteristics of tumors with tumor stroma, which can be applied to advanced 3D cancer organoid models. 

## 5. Conclusions

Advances in the single-cell sequencing technology have provided useful insights for cancer research by revealing high-resolution molecular and genetic landscapes of individual cells or distinct subtypes that contribute to intratumoral heterogeneity. Recently developed advanced organoid models that mimic the TME characteristics, including the vasculature and interactions between tumors and other types of cells, have also improved our understanding of biological diversity and enabled high-throughput in vitro drug testing. These advanced technologies have addressed various research questions related to intratumoral heterogeneity, drug resistance mechanisms, and genetic mutations. Furthermore, single-cell sequencing approaches and organoid technology complement the limitations of each other and together help recapitulate the molecular and biological diversity associated with cancer development, immune response, drug resistance, and therapeutic response. As a result of these studies, precision oncology has been further developed, which improved outcomes and prognoses of cancer patients. Thus, integrated approaches of single-cell transcriptome sequencing and organoid models will provide useful insights and potential preclinical models to identify promising targets for anticancer drugs, facilitating personalized cancer therapy (Figure 2). 

**Table 1 ijms-23-00216-t001:** Single-cell transcriptome sequencing (scRNA-seq) approaches in ovarian cancer and breast cancer.

	Application	Functional Study	Reference
Ovarian cancer	Basic research	Transcriptome expression profiles of individual cells; intratumoral heterogeneity within ovarian cancer and ascites (fibroblast, T cell, B cell, macrophages, dendritic cells)	[40,41,42,43,110]
	Druggable target, translational research	Individual gene expression of immune cells in HGSOC; contribution of JAK/STAT signaling in inflammatory programming; drug screening with cucurbitacin I in vitro and in vivo; identification of grade and origin specific cell populations	[44,47,48]
	Cancer stem cell	Comparison of gene expression profiles between ovarian cancer and embryonic tissues; cell population expressing PEG10 modulates ovarian cancer stemness and drug resistance	[49]
	Drug resistance	Identification of chemo-resistant cell population in HGSOC; the cells express CD44, MYD88, and ALDH1	[53]
Omentum (ovarian cancer)	Druggable target	High T cell infiltration in the omentum in ovarian cancer patients; increase of antitumor response; providing therapeutic targeting	[52]
Breast cancer	Basic research	Single-cell transcriptome profiling of individual cells; clonal evolution; genomic evolution in TNBC; characterization of heterogeneous tumor cells with stromal and immune cells (T cell, B cell, macrophages, CAFs); intratumoral heterogeneity within breast cancer	[64,65,66,67,68]
	Advanced scRNA-seq research	Nanogrid single-nuclear RNA sequencing; heterogenous phenotypic profiles of breast cancer related to angiogenesis, cell proliferation, and cancer stemness	[69]
	Translational research	scRNA-seq analysis using 40 TNBC patients with neoadjuvant anti-PD1; CCR2+ or MMP9+ macrophage and dendritic cells increased T cell expansion; providing therapeutic targeting for synergistic effect with anti-PD1	[80]
	Cancer stem cell, Drug resistance Translational research	CAF-induced Hedgehog ligand promotes chemo-resistant and cancer stem cell population in TNBC; chemotherapy-induced transcriptional reprogramming of resistant signatures; smoothened inhibitors (SMOi) sensitize tumors with docetaxel in vivo; providing a therapeutic target in TNBC	[82,83]

**Table 2 ijms-23-00216-t002:** Organoid models for therapeutic drug sensitivity testing in ovarian and breast cancer.

	Source	Development Efficiency	Features and Use	Reference
Ovarian cancer	56 organoid lines derived from 32 patients	Medium (~65%)	Maintaining CNVs, recurrent mutations and tumor heterogeneity; long-term expansion; providing drug screening platform; in vivo tumorigenicity; sensitive to platinum-based therapy	[91]
	33 organoid lines derived from 22 HGSOC patients	High (80–90%)	Maintaining DNA repair gene mutational status in HGSOC; providing DNA repair profiling and a rapid functional platform for therapeutic sensitivity testing	[92]
	14 organoid lines derived from 3 HGSOC, 1 clear cell, 3endometrioid patients	High (~80%)	Replicating the mutational landscape of the primary tumors; maintaining similar CNVs and *BRCA1* pathogenic variant; sensitivity to PARP inhibitor, olaparib, and platinum drugs	[93]
	14 organoid lines derived from 21 gynecologic tumors	High (~95%)	Retaining features of histology and mutations of original tumors; retention of intra-tumoral heterogeneity; only 1 organoid model has in vivo tumorigenicity; drug response assay using organoid-derived spheroids	[90]
Breast cancer	>100 organoid lines derived from >150 patients	High (>80%)	Matching the histopathology, hormone receptor status, and HER2 status of the parental tumor; generic variations retained after long-term expansion; providing in vitro drug screens; sensitive to drugs (e.g., afatinib and pictilisib) blocking the HER signaling pathway	[94]
	45 biobanked breast organoid cultures	Medium (55–70%) in most subtypes; Low (~40%) in TNBC	Organoids covering all major breast cancer subtypes; providing genetically edited normal breast organoids using CRISPR–Cas9; providing in vitro and in vivo drug screening platform	[97]
	99 organoids derived from 132 samples	Medium (~75%)	Recapitulating the histopathologic and genetic characteristics of parental tumors; in vitro drug sensitivity screening; sensitive to microtubule-targeting drugs	[98]

## Figures and Tables

**Figure 1 ijms-23-00216-f001:**
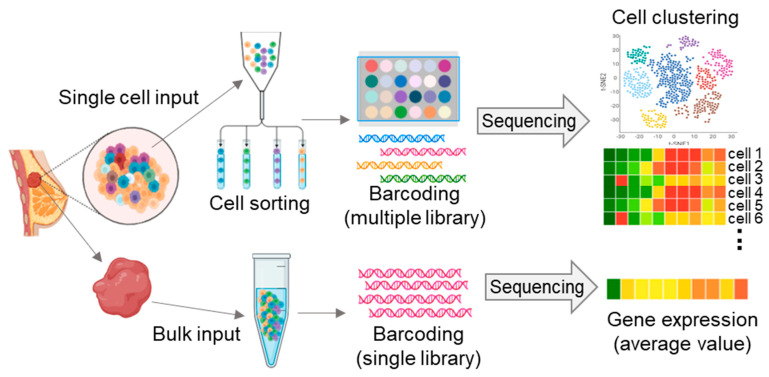
Single-cell RNA sequencing (scRNA-seq) and whole-genome RNA sequencing (bulk RNA sequencing) workflow. scRNA-seq provides unique transcriptome landscape of individual cells (top). Bulk RNA sequencing provides average transcriptome expression of total cells (bottom).

**Figure 2 ijms-23-00216-f002:**
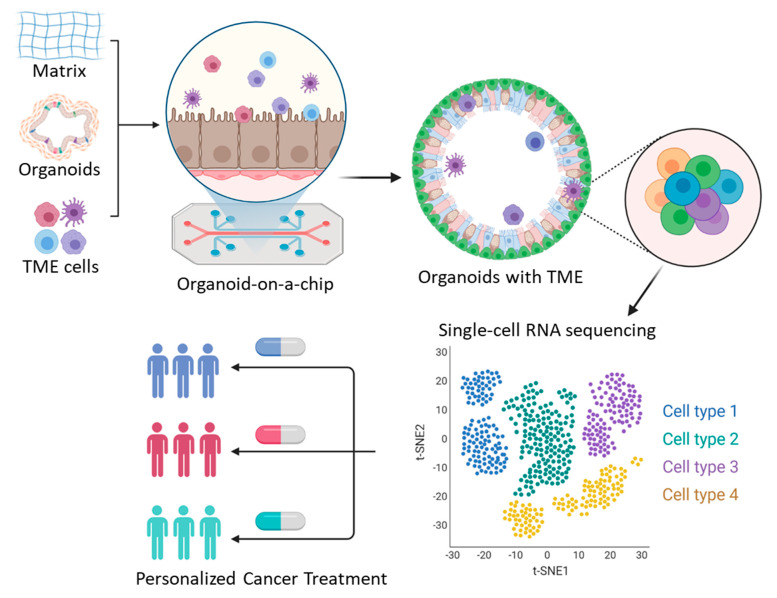
An integrative approach for personalized cancer therapy using advances in tumor organoid and single-cell transcriptome sequencing.

## Data Availability

All data is included in manuscript figures.

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
