# Peer review of "Integration of Genomic Profiling and Organoid Development in Precision Oncology"

_ijms, 2021, doi:10.3390/ijms23010216_

Round 1
Reviewer 1 Report
This is a review article on genomic profiling and organoids in cancer. Overall, the literature cited appears to be thorough, however, I feel that the article is not well structured.
The choice of cancers to include seems somewhat arbitrary, as is their order in the manuscript. Surely it would be better to discuss breast and ovarian cancer in adjacent subsections? Likewise GIST and gastrointestinal. Noticeable was the absence of BRCA mutations in the breast cancer section - why? There are several very long paragraphs that would be better as shorter ones, each clearly conveying a point. For instance in section 2.2, gastric, pancreatic and colorectal are all mentioned at varying times. Similarly in 2.3 with the breast cancer subtypes.
The mutations, and drugs that target them, are not unique to the cancer types. These links/overlaps were not mentioned, e.g. EGFR targeting drugs. Is it the same mutations in a gene for all the different targets? Grouping by gene is just as valid as grouping by cancer type. Either way, the overlap should not be ignored.
Other cancers - focus on glioblastoma and prostate. Are there any other examples?
In the final section on organoids, some abbreviations are undefined (TME, PDX). Much of this section focuses on prostate and glioblastomas which were grouped as "other" in the previous section. Why is there no mention of ovarian or gastrointestinal work in this section? There is a juxtaposition between section 2 & 3. Likewise, the drugs for prostate cancer (everolimus and BKM-120) mentioned in 3.1, are not mentioned in 2.6 - why not?
Link between organoid-on-a-chip and cancer studies is missing? If not yet yet conducted, then please state that this is a future area of research for cancer at the end of the paragraph. In section 4, again ovarian cancer is missing yet it was the first cancer mentioned in section 2.
In the conclusions, there is no mention of "precision oncology", nor its relation to patient treatments and outcomes. How does the conclusion tie in to the overall aim of the review?
Reviewer 2 Report
Although the authors summarized the single-cell analysis method for precision medicine, there was no new idea or insight. Moreover, there was no figure and table. In general, an article paper of the journal of which impact factor is higher than 5 has three or more figures.
Reviewer 3 Report
Below are my findings for the reviewers to consider in their revision:
Introduction
- Line 24: individual genetic characteristics is a broad term and it is not clear if the authors are referring to pharmacogenomics versus a tissue sample extracted from an individual.
- Line 25: I’m not sure the term “heterogenous disease” is the best term to describe cancer. This terms implies a spectrum, which cancer cells and tumors themselves have heterogenous qualities. This should be clarified.
- Line 29: what genetic information is being processed from cancer patients? Is it tumor DNA? Patient immune cell DNA? This should be clarified.
- Line 32: genetic variations and mutations of what lead to increase in cancer? There are many mechanisms that give rise to cancer and this should be flushed out especially leading to the concepts of organoids.
- Lines 42-43: This now introduces a new variant of heterogenous disease. What is being described is really the tumor microenvironment in which the phenotype can change dramatically for a host of reasons. This should be clarified.
- Line 69: Are the authors referring to conventional culture methods? The way this is phrased is unusual.
Section 2
- Generally, I find these subsections lacking in details. While the citations may be correct to illustrate a concept, there is nothing for the reader to truly understand what the studies found or illustrate the findings outlined in the introduction. For example, what has been sequenced and reported as majority/minority mutations in patients? What therapies were tried and what was found (including success and failure rates, subsequent mutations)? From the sequencing data, what were the makeup of the tumors? What about the tumor microenvironment findings? What metabolic or phenotypic changes were uncovered that lend to the complexity of the disease, which then lead to the need for organoid models? I would highly recommend reconstructing these subsections to guide the reader through these findings.
Section 3
- Lines 238-240: describe specifics of what the failures were and where the discrepancies to clinical results were. Show the readers the existing failed links to lend to organoids.
- Similar to section 2, section 3 needs to be flushed out and clearly illustrate what the organoid models successfully proved and elucidated. Did they also drive new treatments for patients? What new drugs were found? What exactly in the tumor environment proves the organoid model is a good proxy for real tissue? Again, I would highly recommend reconstructed these subsections to guide the reader through these findings.
Round 2
Reviewer 2 Report
The authors revised the manuscript with two figures, and it satisfied me. Therefore, I'll recommend this manuscript be accepted.
Reviewer 3 Report
The authors have done a tremendous job improving the manuscript. I have no further comments. Well done.